# Predicting progression-free survival after systemic therapy in advanced head and neck cancer: Bayesian regression and model development

**Paul R Barber[1,2†], Rami Mustapha[3†], Fabian Flores-Borja[4†‡], Giovanna Alfano[3†], Kenrick Ng[1†], Gregory Weitsman[3], Luigi Dolcetti[3], Ali Abdulnabi Suwaidan[3], Felix Wong[3], Jose M Vicencio[1,3], Myria Galazi[1], James W Opzoomer[5], James N Arnold[5], Selvam Thavaraj[6], Shahram Kordasti[7], Jana Doyle[8], Jon Greenberg[8], Magnus T Dillon[9], Kevin J Harrington[9], Martin Forster[1], Anthony CC Coolen[10,11], Tony Ng[1,3,4]\***

[1]UCL Cancer Institute, Paul O'Gorman Building, University College London, London, United Kingdom; [2]Comprehensive Cancer Centre, School of Cancer & Pharmaceutical Sciences, King's College London, London, United Kingdom; [3]Richard Dimbleby Laboratory of Cancer Research, School of Cancer & Pharmaceutical Sciences, King's College London, London, United Kingdom; [4]Breast Cancer Now Research Unit, School of Cancer & Pharmaceutical Sciences, King's College London, London, United Kingdom; [5]Tumor Immunology Group, School of Cancer & Pharmaceutical Sciences, King's College London, London, United Kingdom; [6]Centre for Clinical, Oral & Translational Science, King's College London, London, United Kingdom; [7]Systems Cancer Immunology, School of Cancer & Pharmaceutical Sciences, King's College London, London, United Kingdom; [8]Daiichi Sankyo Incorporated, Newark, United States; [9]The Institute of Cancer Research, London, United Kingdom; [10]Institute for Mathematical and Molecular Biomedicine, King's College London, London, United Kingdom; [11]Saddle Point Science Ltd, London, United Kingdom

**\*For correspondence:**
tony.ng@kcl.ac.uk

†These authors contributed equally to this work

**Present address:** ‡Centre for Oral Immunobiology and Regenerative Medicine, Barts & The London School of Medicine and Dentistry, Queen Mary University of London, London, United Kingdom

## Abstract

**Background:** Advanced head and neck squamous cell carcinoma (HNSCC) is associated with a poor prognosis, and biomarkers that predict response to treatment are highly desirable. The primary aim was to predict progression-free survival (PFS) with a multivariate risk prediction model.

**Methods:** Experimental covariates were derived from blood samples of 56 HNSCC patients which were prospectively obtained within a Phase 2 clinical trial (NCT02633800) at baseline and after the first treatment cycle of combined platinum-based chemotherapy with cetuximab treatment. Clinical and experimental covariates were selected by Bayesian multivariate regression to form risk scores to predict PFS.

**Results:** A 'baseline' and a 'combined' risk prediction model were generated, each of which featuring clinical and experimental covariates. The baseline risk signature has three covariates and was strongly driven by baseline percentage of $CD33^+CD14^+HLADR^{high}$ monocytes. The combined signature has six covariates, also featuring baseline $CD33^+CD14^+HLADR^{high}$ monocytes but is strongly driven by on-treatment relative change of $CD8^+$ central memory T cells percentages. The combined model has a higher predictive power than the baseline model and was successfully validated to predict therapeutic response in an independent cohort of nine patients from an additional

Phase 2 trial (NCT03494322) assessing the addition of avelumab to cetuximab treatment in HNSCC. We identified tissue counterparts for the immune cells driving the models, using imaging mass cytometry, that specifically colocalized at the tissue level and correlated with outcome.

**Conclusions:** This immune-based combined multimodality signature, obtained through longitudinal peripheral blood monitoring and validated in an independent cohort, presents a novel means of predicting response early on during the treatment course.

**Funding:** Daiichi Sankyo Inc, Cancer Research UK, EU IMI2 IMMUCAN, UK Medical Research Council, European Research Council (335326), Merck Serono. Cancer Research Institute, National Institute for Health Research, Guy's and St Thomas' NHS Foundation Trust and The Institute of Cancer Research.

**Clinical trial number:** NCT02633800.

## Editor's evaluation

While immune checkpoint inhibitors (anti-PD-1 targeted agents) are now FDA-approved for the treatment of locally advanced and recurrent or metastatic head and neck cancer, predictive biomarkers are lacking. In this study, the co-authors have developed an algorithm that they conclude predicts the clinical outcome to multimodality immunotherapy. While this machine learning approach is intriguing, prospective validation of the proposed immune-based signature is essential to begin to incorporate such an approach into the clinic.

## Introduction

Recurrent (R) or metastatic (M) head and neck squamous cell carcinoma (HNSCC) is associated with a poor prognosis. For many years, the standard-of-care first-line systemic treatment was the EXTREME regimen, consisting of a platinum-based chemotherapy regimen and cetuximab, an anti-EGFR monoclonal antibody (*Van Cutsem et al., 2009*). The KEYNOTE-048 trial in 2019 changed the treatment paradigm for these patients by incorporating pembrolizumab, an immune checkpoint inhibitor, into the first-line setting (*Burtness et al., 2019*). However, the EXTREME regimen remains a first-line standard-of-care for a substantial number of patients, specifically those with programmed death ligand 1 (PD-L1) negative tumours or those with contraindications to the use of anti-PD1 immunotherapy.

While effective, these regimens are associated with significant toxicities. One of the key challenges for the treating physician is to identify the patients who would benefit from either of these treatment regimens. A predictive biomarker signature for patients with advanced HNSCC will help individualize discussions with patients regarding the risk-benefit balance of treatment and may guide patients who are likely to perform poorly towards alternative therapy regimens or clinical trials.

The absence of predictive biomarkers in this patient cohort represents a significant clinical unmet need. Until the development of PD-L1 as a biomarker for immunotherapy, efforts to generate biomarkers in HNSCC have focused on gene expression profiles, which are dependent on the availability of tumour tissue and are only performed on pre-treatment samples (*Bossi et al., 2016*; *You et al., 2019*). Signatures based on a single biological modality and taken at a single timepoint may be insufficient to predict outcomes, as response to therapy relies on a dynamic interplay between cancer genomics, immune profile, tumour microenvironment, and clinicopathological characteristics of the patient receiving treatment (*Cheerla and Gevaert, 2019*; *Huang et al., 2019*).

Efforts to develop a machine learning model to stratify survival risk by combining genetic and clinicopathological characteristics have revealed some success in advanced oral squamous cell carcinoma (*Tseng et al., 2020*). We hypothesize that a multimodal analysis, taking into account both clinicopathological and laboratory-based biological covariates at different timepoints, would provide better predictive value. Furthermore, by only including covariates that can be obtained from a blood biopsy, patients could be easily screened for discovered biomarkers.

We prospectively collected peripheral blood samples from a Phase 2 trial in R/M HNSCC (NTC02633800) (*Forster et al., 2019*), which utilized cetuximab with platinum therapy as a backbone, and conducted a parallel exploratory analysis with the aim of generating a biomarker signature which would predict outcomes to treatment. We hypothesized that the detailed definition of a broad immune cell signature could contribute to the development of assays employing liquid biopsies to

predict clinical outcomes. We also incorporated the analysis of two circulating microRNAs (miRNAs) from exosomes: *miR-21-5p* and *miR-142-3p*, which have previously demonstrated prognostic and predictive utility (*Summerer et al., 2015*; *Vahabi et al., 2021*). As the trial investigated the efficacy of an anti-ErbB3 antibody, patritumab, administered alongside an anti-EGFR antibody, we simultaneously analysed EGFR-ErbB3 dimerization using Förster resonance energy transfer (FRET) and included it in our analysis.

By extracting information from patient samples at baseline and after the first cycle of treatment within this trial, we aimed to generate a multimodal predictive signature for the systemic therapy based on a novel Bayesian multivariate model. The immune populations driving the risk signature were then prospectively validated in an independent cohort of patients from a Phase 2 trial evaluating avelumab in combination with cetuximab in R/M HNSCC (the EACH trial; NCT03494322). Blood samples were collected at baseline and post first dose of cetuximab before administration of avelumab. Validation was performed against the best overall response, as assessed by iRECIST, showing strong correlation between biomarkers identified in the risk signature and therapeutic response despite changes in treatment regimen. We then used imaging mass cytometry driven by the risk signature to identify correlate changes in systemic immune populations with intra-tumoural immune subsets that colocalize at the tumoural level and appear to be key to response. This risk signature can serve as a non-invasive risk stratification for patients with R/M HNSCC using only peripheral blood, guiding the clinician toward the likelihood of success early during the treatment course.

## Materials and methods
### Study design
The clinical study design of the Phase 2 study (NCT02633800) and its associated exploratory analysis are shown in *Figure 1A*. Eighty-seven patients were enrolled in the clinical trial. Peripheral blood samples were collected at baseline before initiation of treatment (C1) and immediately before the second cycle of treatment (C2). Thirty-one patients were excluded due to incomplete paired biological datasets, leaving 56 patients for analysis. Amongst these patients, there was no difference in PFS as demonstrated by Kaplan-Meier survival curve analysis (*Figure 1—figure supplement 1*) regardless of whether the patients received patritumab, which reflected the results published in the clinical trial. The baseline clinical characteristics of these 56 patients are shown in *Supplementary file 1* together with a comparison that shows there is no significant difference between the clinicopathological characteristics pertaining to the discovery cohort and the whole cohort of study.

PBMC samples were analysed using flow cytometry to generate unique immunological subpopulations. Exosomes were extracted from the serum and analysed for EGFR-ErbB3 dimerization and *miRNA-21-5p* and *miRNA-142-3p* (*Figure 1B*). These analyses yielded a total of 29 unique biological covariates. Each covariate was obtained in pairs (C1 and C2), generating a total of 58 laboratory-based covariates for the multivariate analysis (*Figure 1B*). To mitigate individual baseline variations between patients, the biological data obtained from the C2 timepoint was evaluated as relative change with respect to the baseline value of the same covariate at C1 (in the form of log2-fold change [lfc] of the variable of interest) instead of absolute values of those parameters. A list of the laboratory-based and clinical covariates is provided in *Supplementary file 2*.

The baseline clinical characteristics, as well as value of the laboratory-based covariates at baseline and after one cycle of treatment, did not significantly differ between the placebo and patritumab cohorts (*Supplementary file 3*). Therefore, in this exploratory analysis, samples from both the control and investigational arms were analysed together. The effect of adding the investigational product, patritumab, on progression-free survival (PFS) was evaluated by including it as an independent clinical covariate, denoted as 'Drug', in our multivariate analysis.

Written informed consent was obtained. Approval was obtained from ethics committees (Research Ethics Committee reference: 15/LO/1670).

The design of the EACH trial (NCT03494322) that was used to validate the signature is shown in *Figure 5—figure supplement 1A*. The aim of the EACH trial was to evaluate the safety and antitumour activity of avelumab and cetuximab in R/M squamous cell carcinomas. Sixteen patients were enrolled in the study. Four patients were excluded due to lack of blood samples at both timepoints. One patient was excluded due to lack of outcome data. Two patients only had a baseline biopsy.

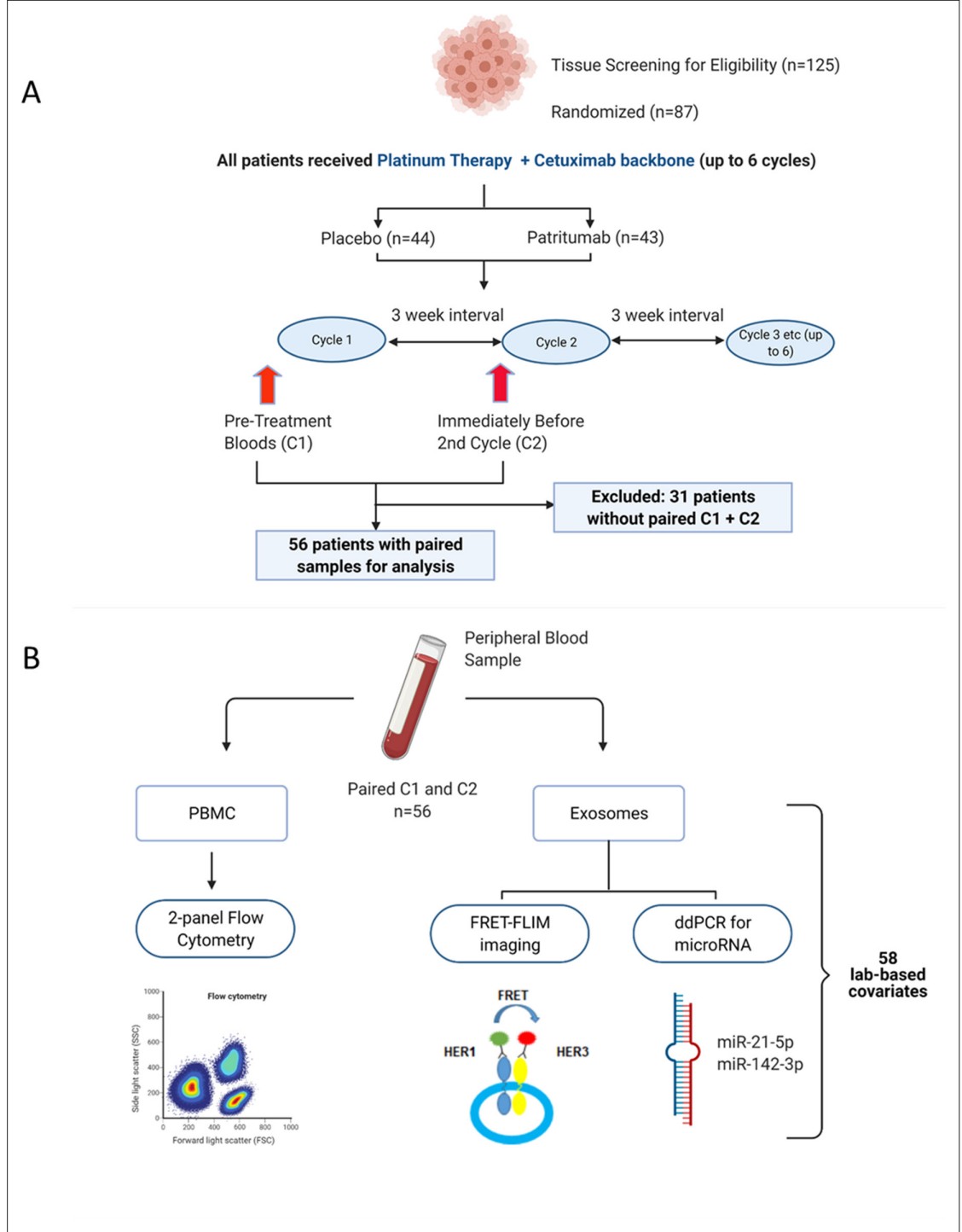

**Figure 1.** Peripheral blood samples from the clinical trial were prospectively analysed using a multimodality platform. (**A**) Schematic of clinical trial design and timepoints at which peripheral blood was obtained. (**B**) Fifty-six (n=56) paired blood samples, obtained pre-treatment (C1) and after one cycle of treatment (C2) were subjected to flow cytometry, Förster resonance energy transfer-fluorescence lifetime imaging microscopy (FRET-FLIM) imaging, and droplet digital polymerase chain reaction (ddPCR) analysis.

The online version of this article includes the following figure supplement(s) for figure 1:

**Figure supplement 1.** Kaplan-Meier curve of progression-free survival in study cohort.

**Figure supplement 2.** Gating strategies for definition of peripheral blood immune populations.

Written informed consent was obtained from all patients. Study approval was obtained from ethics committees (Research Ethics Committee reference: 18/LO/0021). PBMCs underwent a quality control step to check viability which resulted in the exclusion of one of the patients (*Figure 5—figure supplement 1B*). One patient died before their first response evaluation scan and was therefore classified as having progressive disease as the cause of death was identified to be the illness.

## Statistical analysis

To examine whether the covariates indicated different prognostic outcomes, we built a model for predicting PFS. Using Bayesian multivariate proportional hazards regression, covariates were ranked and selected by predictive importance (*Grigoriadis et al., 2018*). We derived two models using separate datasets: firstly, a baseline predictive model containing a dataset of 42 baseline covariates (29 laboratory parameters at baseline (C1) and 13 clinical characteristics). The second, a combined predictive model, consists of 71 covariates, that is, the 42 baseline covariates and a further 29 derived from the on-treatment change of the value of a lab-based parameter relative to its value in the same patient at baseline denoted as lfc of that variable.

The relative efficiency of the predictive model was assessed by using C-index (a metric proposed by *Harrell et al., 1984*, to evaluate the accuracy of predictions made by an algorithm) and rank correlation of the signature-generated risk scores with survival time. The number of significant covariates in each prediction signature was determined with the aim of avoiding overfitting of the signature to the study data using the 'batch regression' option of the Saddle Point Signature software (Saddle Point Science Ltd., London, UK), according to methods that were previously published (*Barber et al., 2020*; *Shalabi et al., 2018*). Systematic iterative covariate rejection and cross-validation (5000 iterations) allowed for the selection of an optimal covariate set to avoid overfitting though inclusion of too many covariates. The optimal set can be chosen in two ways, either based on the peak prediction performance of cross-validation or the more stringent method that equally penalizes validation performance and overfitting (defined as the deviation between training and validation performance). All signatures presented were chosen using the more stringent criterion and data for all covariates is also presented for the purposes of identifying covariates that may be important but do not quite meet the criterion. The regression included covariates representing the missingness of the data to account for the possibility that patient or sample selection/rejection (for any reason) is biased with respect to outcome and therefore could be informative.

Covariates were normalized to zero mean and unit standard deviation such that the importance and significance of covariates can be judged by their assigned beta value (β) in the proportional hazards model, and corresponding hazard ratio (HR) equal to $e^{2\beta}$. A negative β value reflects a lower risk of developing an event. The Saddle Point Signature software additionally judges the performance of similar randomized data, which most often has β values around zero and within a critical range, such that any real covariate that has a β value outside this critical range can be judged to be performing significantly better than randomized data. For standard correlation tests between numeric covariates, Pearson correlation was used and denoted as r. For correlation tests against categorical outcomes, or where correlations may not be linear (e.g. PFS), Kendall rank correlation was used which was denoted as $\tau$. Kaplan-Meier curves and log rank tests were generated using the R survival package. Correlations were performed with the R cor.test function. C-index values were calculated using the Hmisc R package.

Validation of the signatures was performed by calculating the risk score for each patient in the validation trial (EACH) using the combined risk signature covariate weights and performing a correlation test versus the iRECIST outcome for that trial. The covariates were obtained from flow cytometry analysis on the PBMCs of the validation cohort and patient age at registration. As detailed tumour site information was not available for the validation cohort, these covariates were replaced in the signature by the average values from the discovery cohort.

## Flow cytometry

Frozen PBMC samples were thawed and stained with Fixable viability dye (Yellow Live/Dead, Thermo Fisher Scientific) followed by two different panels of membrane markers. Two different panels were used, a simple one for the discovery cohort and a more thorough panel for the validation cohort (full list of both antibody panels in *Supplementary file 4* and *Supplementary file 5*). These two panels

allow definition of immune cell populations as described in *Figure 1—figure supplement 2*. Patients' samples and corresponding Fluorescence Minus One (FMO) Controls were acquired in a Fortessa II flow cytometer (BD, Berkshire, UK) and analysed with FlowJo software (Tree Star).

## Isolation of serum exosomes

Exosomes were prepared using an optimized centrifugation method (*Monypenny et al., 2018*). Diluted serum was centrifuged at 300 × *g* for 10 min to remove cell debris, 5000× *g* for 20 min to remove large vesicles and membrane fragments, and 12,200× *g* for 30 min to deplete microvesicles. This was followed by 100,000 × *g* ultracentrifugation for 120 min at 4°C to pellet exosomes with a TLA-55 rotor (Beckman Coulter). After a second 100,000 × *g* ultracentrifugation for 60 min, the resulting pellets were washed and resuspended in PBS. Purified exosomal fractions were diluted and used for nanoparticle tracking analysis using a Nanosight LM-14 system.

## RNA extraction and miRNA expression analysis

RNA from cancer patients' serum exosomes was extracted using the TRIzol Plus RNA Purification Kit (Thermo Fisher, UK) according to the manufacturer's instructions. Quantification of gene expression in circulating exosomes was performed by droplet digital polymerase chain reaction (ddPCR) (Bio-Rad QX100 system). Normalization of the RNA, between cycle 1 and cycle 2 therapy of each patient, was performed using the expression levels of the housekeeping gene *18S* (Assay ID, Hs99999901_s1). For each sample, equal volume of RNA was used as template and cDNA synthesis performed using the SuperScript VILO MasterMix (Thermo Fisher, UK) according to the manufacturer's instructions. MicroRNAs were reverse-transcribed individually using the TaqMan MicroRNA Reverse Transcription Kit (Thermo Fisher, UK). For each sample, the normalized amount of RNA was reverse-transcribed in a 15 µl reaction using the standard protocol and primers specific for each miRNA: *miR-21-5p* (assay ID, 000397), *miR-142-3p* (assay ID, 000464). Then, 7.5 µl of cDNA was added to a 20 µl reaction containing 12.5 µl 2× ddPCR Supermix for Probes (Bio-Rad) and 1 µl 20× TaqMan miRNA PCR primer probe set; each reaction was carried out in duplicate. Thermo cycling conditions were as follows: 95°C for 10 min, then 50 cycles of 95°C for 10 s and 61°C for 30 s and a final inactivation step at 98°C for 12 min. PCR products were analysed using the QuantaSoft Software (Bio-Rad).

## ErbB3-EGFR dimer quantification in exosomes

Exosomes were imaged on an 'Open' fluorescence lifetime imaging microscopy (FLIM) system (*Barber et al., 2013*). Analysis was performed with the TRI2 software (v2.7.8.9, CRUK/MRC Oxford Institute for Radiation Oncology, Oxford) as described previously (*Barber et al., 2009*; *Rowley et al., 2016*). Interfering effects of autofluorescence were minimized with a lifetime filtering algorithm and the FRET efficiency value for each patient calculated by: FRET $= 1 - \frac{\tau_{DA}}{\tau_D}$ , where $\tau_D$ and $\tau_{DA}$ are the average lifetime of Alexa Fluor 546 in the matching donor (D) and donor-acceptor (DA) images.

## Imaging mass cytometry

Formalin-fixed paraffin-embedded (FFPE) histological slides were stained with a panel of metal conjugated antibodies (full list of antibodies listed in *Supplementary file 6*).

In brief, antigen retrieval was performed on a Ventana Bench Mark Ultra with CC1 buffer (Roche, 950-224). Slides were blocked for 1 hr at room temperature in 5% BSA, 5 mg/ml human IgG in PBS, and stained overnight at 4°C in 4% BSA, PBS. DNA counterstain was performed with Iridium (Fluidigm, 201192B) 125 nM in PBS for 30 min at room temperature.

Ablation and data acquisition of multiple regions of interest per tissue section were performed on a Fluidigm Hyperion. Imaging analysis was performed using the following R packages: RandomForest for classification and regression, Raster and SF for image manipulation and segmentation. Scripts are available in the GitHub link provided in the Data availability section of this manuscript.

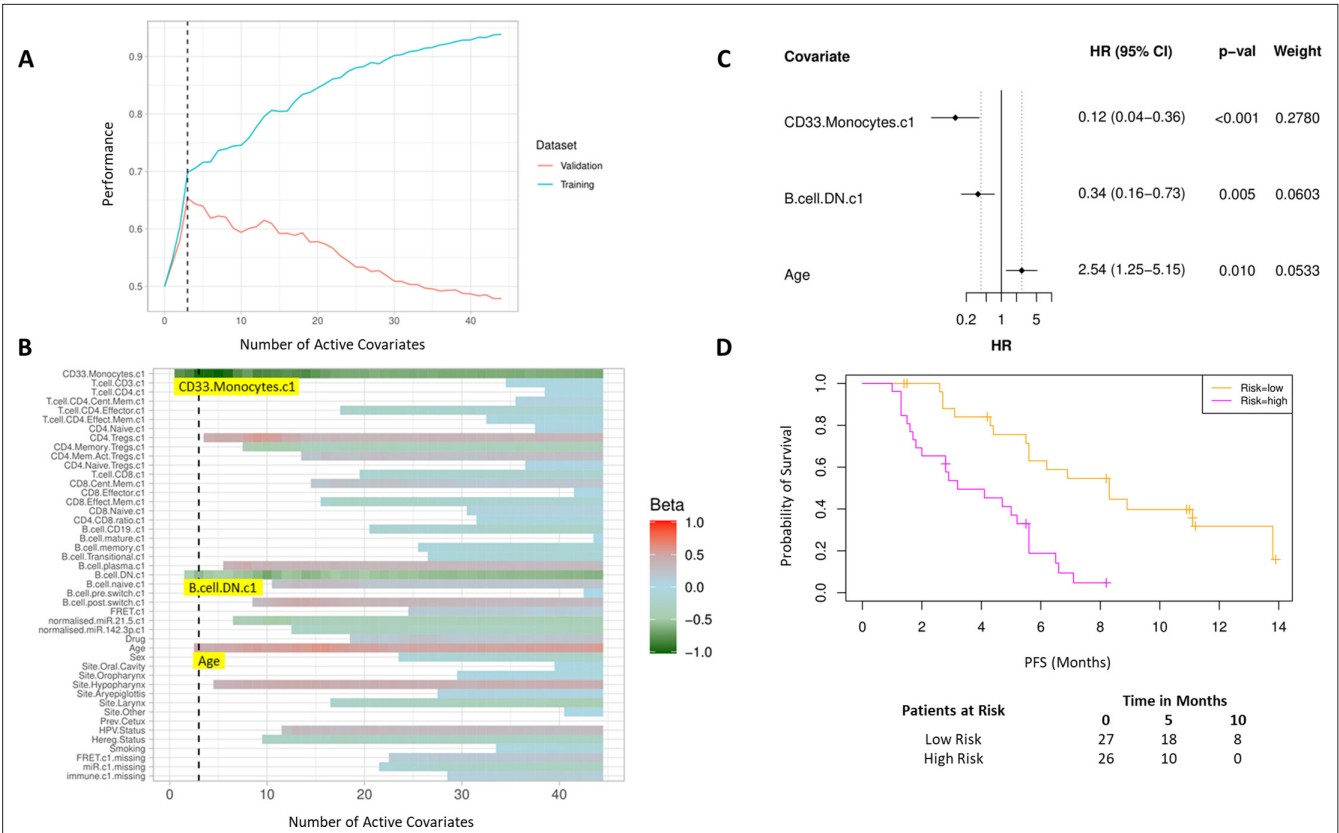

**Figure 2.** High baseline CD33+CD14+ monocytes and double negative B cells predict progression-free survival (PFS). (**A**) Covariates were ranked for importance and selected by a proportional hazards regression model with cross-validation. (**B**) Proportional hazards regression revealed three covariates which exceed the beta critical value – CD33+CD14+ monocytes, double negative B cells, and age. (**C**) Forest plot of the three covariates within PFS risk score with dotted line indicating the range, around 1, of typical random covariates. (**D**) PFS risk signature performance, low risk score (n=27) and high-risk score (n=26). Log rank p-value = 6.0e-5, with numbers at risk demonstrated under Kaplan-Meier curve. The multivariate analysis resulted in risk signatures that are linear combinations of weighted covariates. Their ability to predict outcome is demonstrated with data split by signature value.

## Results

### The model with baseline covariates reveal immune subpopulations and age predict PFS

Bayesian multivariate proportional hazards regression was performed on the 42 covariates derived at baseline (C1) and PFS outcome. We utilized the stringent selection criteria based on a proportional hazards regression model to minimize overfitting based on the cross-validation performance (*Figure 2A*). This revealed two baseline immune subpopulations with a β value which exceeded the critical β value threshold, that is, CD14+CD16+CD33+CD11b+ monocytes thereafter referred to as CD33+CD14+ monocytes according to previous nomenclature (*Cravens et al., 2007*) and double negative (CD27-IgD-) B cells (DN B cells), as well as one clinical covariate – age (*Figure 2B*). Missingness covariates were included in this analysis and did not affect the outcome of the signature.

Evaluation of the individual β values reveal that baseline CD33+CD14+ monocytes and DN B cells have a β value of –1.05 and –0.53, respectively, and hence a higher baseline value of both populations is predictive of better PFS. Age, with a β value of 0.47, is associated with poorer PFS. The HRs of the individual covariates are depicted in *Figure 2C*. The baseline risk scores correlated strongly with PFS (C-index=0.60, $\tau$ =−0.33, p=0.0005). The risk score equation is given in *Supplementary file 7*.

The risk scores generated from this signature were split at the median value to generate low-risk and high-risk cohorts (*Figure 2D*). The median PFS of the low-risk and high-risk cohorts are 8.3 and 3.6 months, respectively (log rank p-value = 6.0e-5).

## Incorporating laboratory-based covariates after one cycle of treatment improves ability to predict PFS benefit

We subsequently evaluated if the incorporation of early laboratory-based changes into the signature improves its predictive ability. A separate predictive model incorporating an additional 29 new covariates, that is, relative on treatment changes in laboratory-based parameters at cycle 2 with respect to the value at baseline (C1).

As before, we used a proportional hazards regression to determine a set of variables which predict PFS. A total of six covariates were identified – three immune subpopulations with negative β values and hence associated with better survival, that is, baseline CD33$^+$CD14$^+$ monocytes, baseline CD4$^+$ memory regulatory T cells (HLA-DR$^-$CD45RO$^+$Tregs), and an increase in CD8$^+$ effector memory T cells (CD45RO$^+$CCR7$^-$). An increase in two subpopulations, CD8$^+$ central memory T cells (CD45RO$^+$CCR7$^+$) and CD3 T cells, was associated with inferior PFS. The hypopharyngeal primary tumour site was also associated with a poorer PFS (*Figure 3A and B*).

A multivariate analysis employing linear combinations of these six weighted covariates generated a risk signature. The combined risk scores exhibited a stronger correlation with PFS than the baseline risk scores (C-index=0.69, $\tau$ =−0.49, p=7e-7). Their ability to predict outcome is demonstrated with data split by risk score, shown in *Figure 3C*. In this combined predictive signature, the median PFS of the low-risk and high-risk cohorts are 6.8 and 3.6 months, respectively (log rank p-value 0.004) (*Figure 3C*). The risk score equation is given in *Supplementary file 7*.

## EGFR-ErbB3 FRET on exosomes may contribute to predictive signature

While the combined predictive signature comprised predominantly of immunological parameters, there is a suggestion that FRET difference may carry a degree of predictive value. In *Figure 3A* (fourth covariate from the bottom), the difference in EGFR-ErbB3 FRET (FRET.delta) was associated with a negative β value which suggests a better PFS. However, the stringency that we have applied to optimal covariate selection means that this covariate fell marginally short of featuring in the eventual predictive signature. Nonetheless, this is the first time that this assay, (explained in *Figure 4B*), has been used within the context of a randomized controlled trial in exosomes and the suggested predictive value of the dimer warrants some discussion. *Figure 4A* displays intensity images and donor lifetime map of exosomes labelled with anti-EGFR and anti-ErbB3 antibodies.

By dividing the patients with available FRET values by the median FRET.delta (n=43), there was a suggestion that patients with a high FRET.delta exhibited a better PFS than patients with a low FRET.delta. This difference was not statistically significant (log rank p-value = 0.2) (*Figure 4C*). The predictive capacity of this univariate is limited ($\tau$ =−0.13, p=0.2, C-index=0.586). Nonetheless, these results suggest a trend within a small patient cohort and can be explored in future prospective studies. While none of the remaining exosome derived perimeters correlated with PFS, miRNA signatures have been implicated as a useful classifier for myeloid cell subsets (*Bronte et al., 2016*). By correlating the miRNA changes in our study with this monocytic subpopulation, a significant correlation was identified between the log fold changes of *miR-21-5p* with the corresponding log fold changes of CD33$^+$CD14$^+$ monocytes (r=0.43, p=0.02, *Figure 4—figure supplement 1*).

## Validation of the risk signature in an independent cohort

The validation cohort of 16 patients was obtained from the EACH trial (NCT03494322) which evaluated the combination of avelumab with cetuximab in HNSCC patients. As the risk signature consisted predominantly of immune subpopulations, we successfully obtained PBMCs from eight patients at the pre-treatment timepoint and after one cycle of cetuximab hence matching the timepoints of the original cohort. PFS data in the validation cohort was not entirely available at the time of analysis, hence we used best objective response (BOR) using iRECIST criteria to validate the combined risk signature. We confirmed that in the original test cohort, patient PFS data strongly and inversely correlated with BOR ($\tau$ =−0.52, p=3e-6) (*Figure 5A*). Age and site information were not available for the validation cohort and were replaced in the risk signatures by average values from the discovery cohort.

The combined risk signature strongly correlated with a poorer treatment outcome ($\tau$ =0.73, p=0.02) (*Figure 5B*). Separately, each of the variates showed similar trends in the validation cohort to those observed in the test cohort yet only two key immune populations showed interesting strong correlations (*Figure 5—figure supplement 2*). High pre-treatment levels of CD33$^+$CD14$^+$monocytes

Figure 3

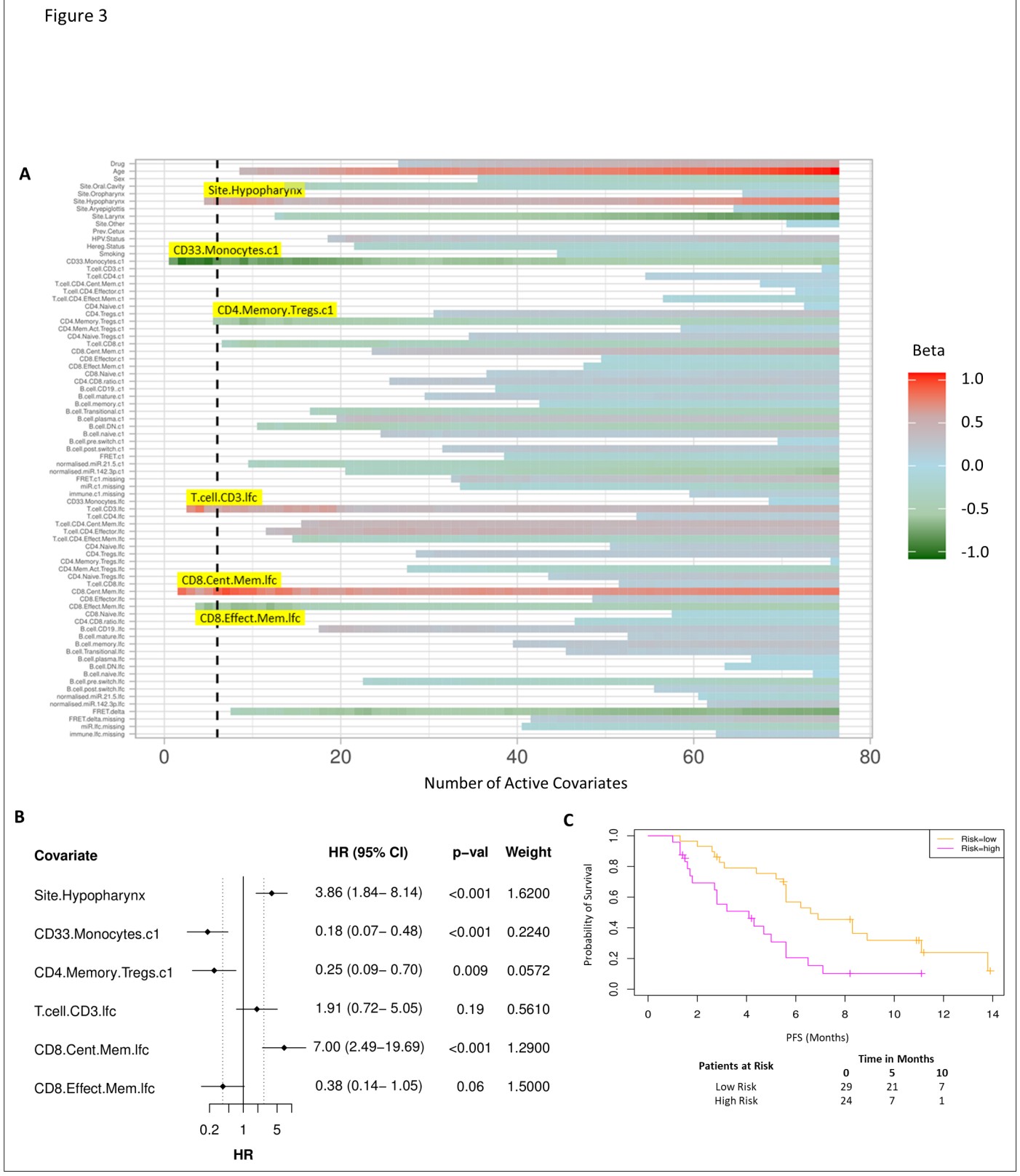

**Figure 3.** Model incorporating laboratory changes after one cycle of treatment exhibit improved predictive value. (**A**) Proportional hazards regression revealed five immune subpopulations which exceed the beta critical value – baseline CD33+CD14+ monocytes, baseline CD4+ memory regulatory T cells, log2-fold change (lfc) of CD8+ effector memory T cells, lfc of CD8+ central memory T cells, and lfc of CD3+ T cells. The primary tumour site of hypopharynx also featured in the signature. A negative beta value is associated with lower risk score and hence better progression-free survival. (**B**)

*Figure 3 continued on next page*

Figure 3 continued

Forest plot of the three covariates within progression-free survival risk score. (**C**) Progression-free survival risk signature performance, low risk score (n=29) and high-risk score (n=24). Log rank p-value = 0.004, with numbers at risk demonstrated under Kaplan-Meier curve.

significantly correlated with a worse disease outcome in all patients except for a single outlier who exhibited a complete response to treatment. A post-treatment increase in CD8[+] central memory cells (lfcCD8CM) also correlated with a poorer treatment outcome. Interestingly, both covariates were the strongest drivers of the risk signature in the discovery cohort with the baseline level of CD33[+]CD14[+]monocytes being a key covariate in both signatures. Given this result we assessed their ability to predict PFS as univariates in the original cohort. *Figure 5C* shows Kaplan-Meier curves of PFS by median of each of these covariates. We observed a modest split that reached significance only in the case of baseline CD33[+]CD14[+] monocyte (log rank p-value = 0.03) and not in the case of lfcCD8CM cells (log rank p-value = 0.1). lfcCD8CM more strongly predicted PFS (C-index=0.74, $\tau$ =−0.36, p=0.01) than base line CD33[+]CD14[+] monocytes (C-index=0.39, $\tau$ =0.22, p=0.07). Due to the consistency with which the CD33[+]CD14[+] monocyte population appeared across our study, we wanted to further characterize this population to determine its phenotype hence we expanded the antibody

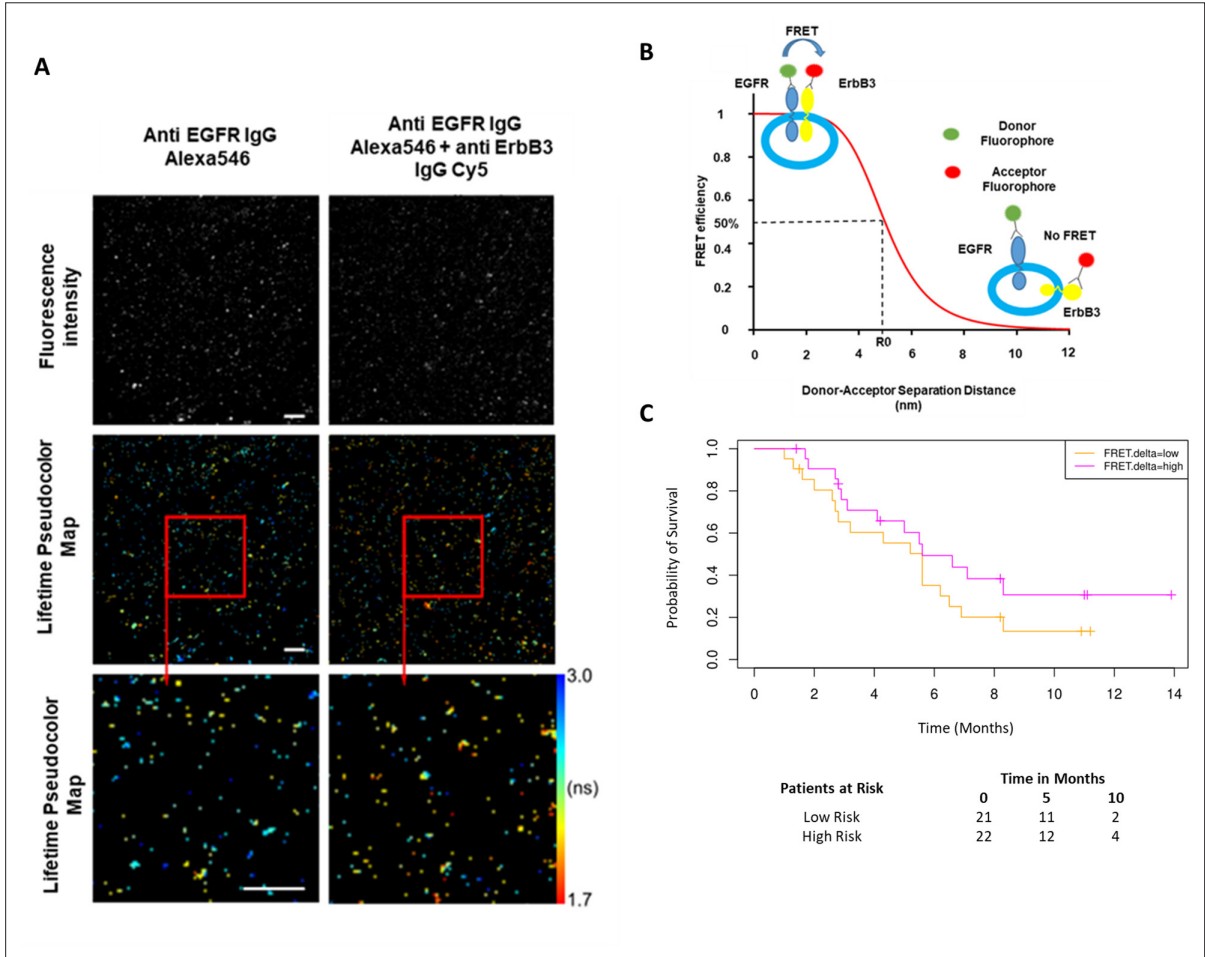

**Figure 4.** Förster resonance energy transfer/fluorescence lifetime imaging microscopy (FRET/FLIM) assay of circulating exosomes extracted from patients. (**A**) Time-resolved fluorescence intensity images and donor lifetime map of exosomes labelled with Anti-EGFR-IgG-Alexa 546 and Anti-ErbB3-IgG-Cy5 extracellular antibodies. (**B**) Schematic illustration of the fluorescent labelling geometry on exosomes and distance dependence of FRET efficiency. (**C**) Progression-free survival of subpopulations divided by median FRET difference, FRET.delta low (n=21) and FRET.delta high (n=22). Log rank p-value = 0.2, with numbers at risk demonstrated under Kaplan-Meier curve.

The online version of this article includes the following figure supplement(s) for figure 4:

**Figure supplement 1.** Correlation between miRNA-21 fold change and CD33[+]CD14[+]monocyte fold change after one cycle of treatment.

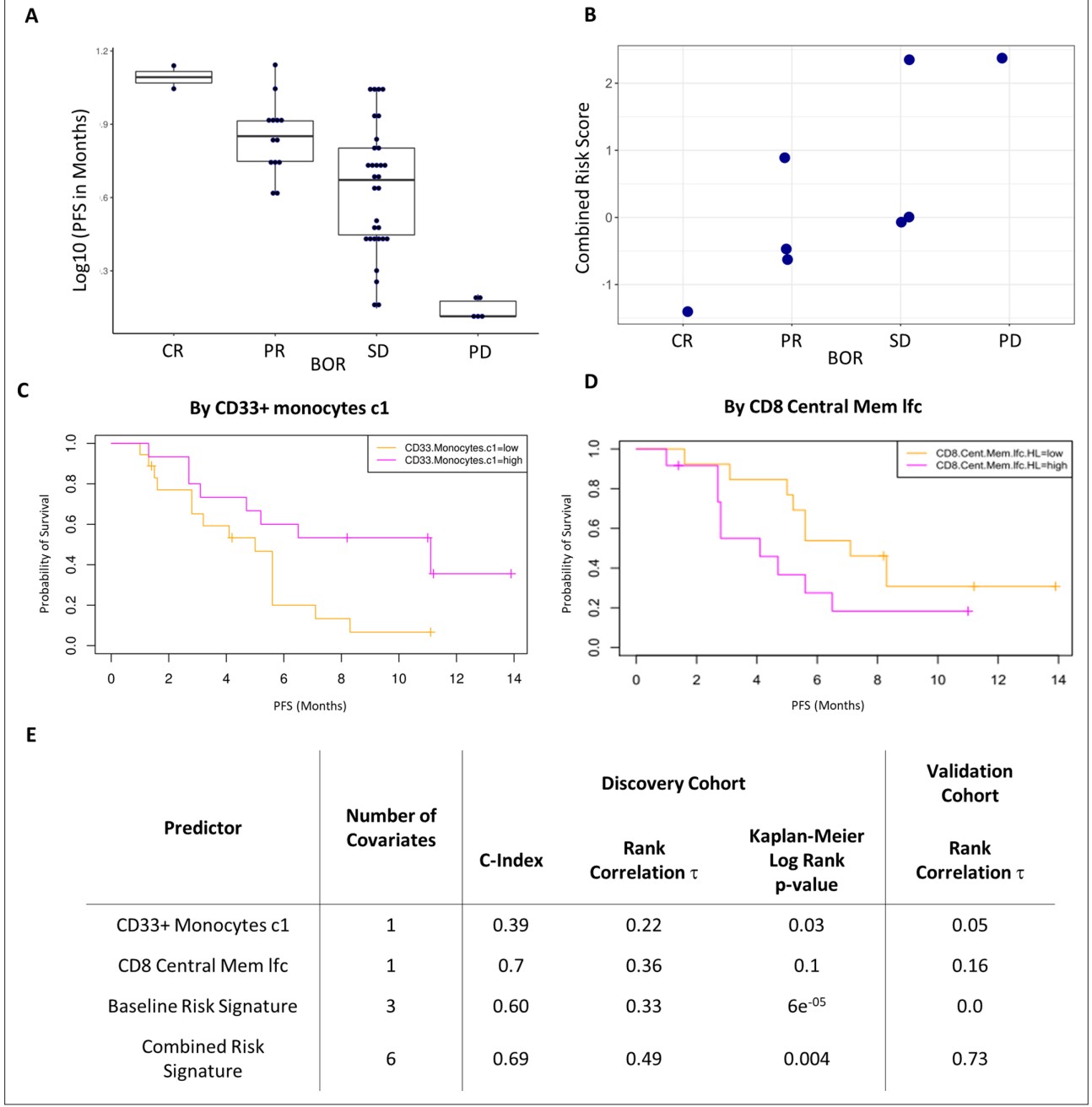

**Figure 5.** Validation of the combined risk score in the EACH cohort. (**A**) Correlation between progression-free survival (PFS) and best objective response (BOR) in the original cohort ($\tau$ =−0.52, p=3e-6). (**B**) Correlation between the combined risk signature and the BOR of the patients in the validation cohort ($\tau$ =0.74, p=0.02). (**C**) Kaplan-Meier curves of PFS split by median percentage of baseline $CD33^+CD14^+$ value. (**D**) Kaplan-Meier curves of PFS split by median value of on treatment fold change of $CD8^+$ central memory T cells relative to baseline. lfc: log2-fold change. (**E**) Table summarizes C-index, rank correlation, and log rank p-value based on type and number of covariates in both cohorts.

The online version of this article includes the following figure supplement(s) for figure 5:

**Figure supplement 1.** The validation EACH trial.

**Figure supplement 2.** Correlation between each of the significant covariates derived in the combined risk signature with treatment outcome in the EACH cohort.

**Figure supplement 3.** Gating strategy for further characterization of $CD33^+CD14^+$ monocytic population using new patient cohort.

panel for the validation set and determined that these CD14[+]CD16[+]CD33[+]CD11b[+] monocytes also express high levels of HLA-DR (*Figure 5—figure supplement 3*).

## Imaging mass cytometry of tissue reveals correlation of CD33[+]HLADR[high] myeloid cell with tissue CD74[+] macrophages interacting with tissue CD8[+] memory T cells

Having established that two immune subsets in peripheral blood predict therapeutic response within a multivariate signature, we subsequently explored the relationship between the immune findings in peripheral blood with tumour infiltrating leukocytes (TILs). We obtained sufficient tissue from the biopsy at trial enrolment for in-depth profiling by imaging mass cytometry from nine patients.

Standard FFPE samples from these patients were stained using a custom immune focused imaging mass cytometry panel (*Supplementary file 6*). The cells were segmented and then classified into superpopulations using a trained machine learning approach. These segmented cells were divided into subpopulations by unsupervised clustering using uniform manifold approximation and projection (*Figure 6A*). The distribution of the superpopulations per patient is shown in *Figure 6B*, yet none of which correlated with PFS (*Figure 6—figure supplement 1*). The phenotypes of the subclusters are shown by heat maps (*Figure 6C*). The key populations were then manually annotated and, in some cases, merged to compensate for over clustering and analysed for their correlation with PFS as well as peripheral immune populations. The complete list of these correlations can be seen in *Figure 6—figure supplement 1*. We focused on the two main populations highlighted by the validation cohort, being CD33[+]CD14[+]monocytes and CD8CM to understand how they are represented at the tumoural level. The peripheral intermediate monocyte population strongly correlated with a tissue CD14[+]CD33[+]CD74[+]CD68[+] macrophage population while the CD8CM inversely correlated with CD45RO[+]CD27[+]CD8[+] tissue resident memory T cell (CD8Trm) population. Both of those populations positively correlated with PFS. Interestingly, the peripheral CD33[+] monocyte population strongly correlated with CD8Trm (*Figure 6D*). To understand whether this correlation resulted in cellular levels of interactions, we assessed the average number of cell-to-cell contact that CD8Trms had with each of the clustered subpopulations. We found these cells were significantly more likely to be in contact with a CD74[+] macrophage than any other identified population suggesting this interaction is biologically driven and not random (*Figure 6E*).

## Discussion

There is a clinical unmet need to identify predictive biomarkers for treatment in head and neck cancer. Gene expression profiling has revealed promising initial results in this domain but have been limited to HPV-positive HNSCC which inherently have better prognoses. The ratio of neutrophils to lymphocytes (NLR) in the peripheral blood of HNSCC patients prior to treatment has been extensively investigated for its ability to predict disease outcome. Two recent meta-analyses on the subject have identified that a high pre-treatment NLR correlated with worse OS with respective HRs of 1.69 (95% CI: 1.47–1.93; p<0.001) and HR of 1.78 (95% CI: 1.53–2.07; p<0.0001) (*Mascarella et al., 2018*; *Takenaka et al., 2018*). Using our combined risk signature, we were able to predict disease outcome more accurately across two separate treatment modalities. Other recent biomarker research has focused on predicting response to immune checkpoint blockade by analysing tissue-based biomarkers, such as PD-L1 expression levels, but when used in isolation have not been sufficiently predictive at identifying patients who would benefit (*Burtness et al., 2019*).

While unimodal biomarkers may offer some predictive value, the biology of HNSCC and likelihood of response to treatment is likely to be dictated by an interplay between tumour immunity, genomic signatures, and a host of clinicopathological characteristics which can only be assessed by a multivariate analysis. There has been an increased interest in peripheral blood-based liquid biopsies in recent years, particularly in the context of PBMC analysis (*Nixon et al., 2019*). The ability to extract predictive biomarkers from a blood-based liquid biopsy mitigates certain limitations posed by tissue biopsies – particularly the tissue accessibility, technical expertise to obtain the biopsy, patient frailty, and the amount of tissue available. The ease of obtaining liquid biopsies also facilitates longitudinal monitoring of response to treatment. Moreover, PBMCs offer a much more systemic snapshot of the immune response which overcomes intra-tumoural heterogeneity that results in non-uniform

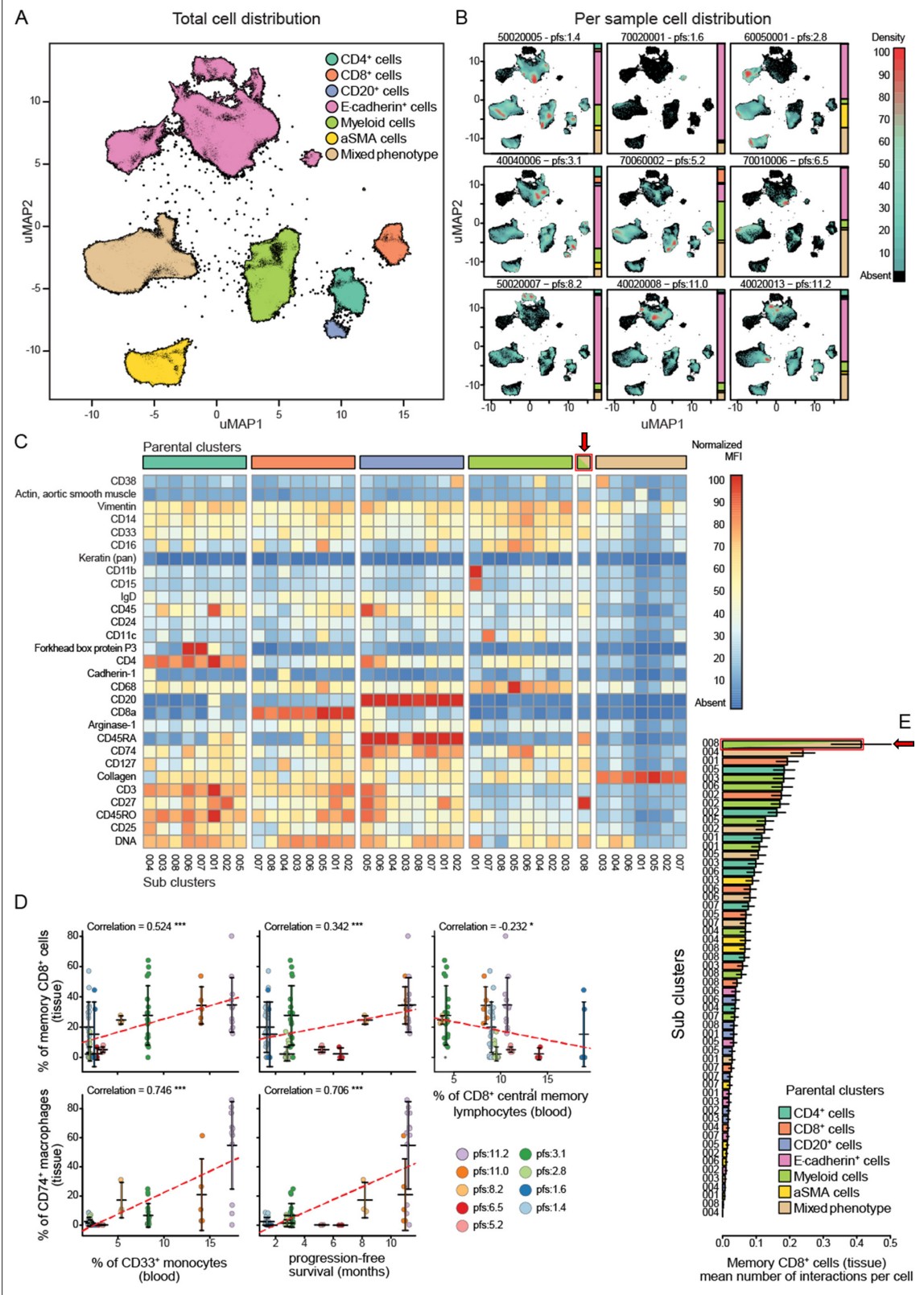

**Figure 6.** Imaging mass cytometry analysis on tissue sections from nine patients. (**A**) Uniform manifold approximation and projections (UMAP) of the segmented cells from the combined patient sections showing their distribution across the superpopulations. (**B**) Relative abundance of the cells across the patient samples arranged in increasing order of progression-free survival (PFS). (**C**) Marker expression pattern across each of the clustered subpopulations. (**D**) Correlations between peripheral PBMC populations and tissue immune populations or between tissue immune populations and

*Figure 6 continued on next page*

*Figure 6 continued*

PFS. (**E**) Average number of interactions between the CD8[+] tissue resident memory T cells and each of the clustered subpopulations. Arrow indicates the population of tissue macrophage that corresponds to peripheral CD33[+]CD14[+]monocyte.

The online version of this article includes the following figure supplement(s) for figure 6:

**Figure supplement 1.** Correlations of the clustered superpopulations with progression-free survival (PFS).

distribution of biomarkers. The potential of PBMCs to unravel the tumoural immune landscape is slowly being realized, thanks to advancements in the fields of high-dimensional flow cytometry, single-cell RNA sequencing, and cytometry by time of flight. However, such techniques generate an excessive number of variables which require powerful statistical analysis tools to overcome pitfalls such as overfitting (*Shalabi et al., 2018*).

To our knowledge, the current study is one of the first examples of integrating multiple biological covariates derived from peripheral blood and patient clinical data to generate a signature which predicts treatment response and that has been validated across multiple therapeutic modalities.

By employing cross-validation iterations to estimate training and validation errors, implementing advanced overfitting correlation protocols, using built-in corrections for informative data missingness, and probabilistic covariate removal, we were able to derive a robust optimal covariate set which correlates with PFS. This combination of analyses has been shown to produce robust signatures that do generalize to uncaptured data (*Barber et al., 2020*).

The biological components of the predictive model warrant discussion. The only clinical covariate to feature in the combined risk signature was the hypopharyngeal SCC sub-site, which was adversely correlated with PFS. This corroborates previous findings that the 5-year relative survival of patients with hypopharyngeal SCC is consistently the worst amongst different anatomical HNSCC sub-sites (*Gatta et al., 2017*; *Machiels et al., 2020*). The propensity of hypopharyngeal tumours to present at the de novo advanced stage (*Cadoni et al., 2017*) and the density of submucosal lymphatics in this anatomical region translates into these patients inherently performing worse – lending support to the robust nature of our predictive signature. The notable absence of patritumab (denoted as 'Drug') in our predictive signatures is also consistent with the outcome of the Phase 2 clinical trial where the addition of this investigational medicinal product did not produce any benefit to PFS (*Forster et al., 2019*).

Another key discovery in this study is the significance of analysis of sequential samples even at early stages of therapy. This, we believe, is important for the analysis of lab-based biological variables. There is a high level of inter-individual heterogeneity in the relative abundance of PBMC subpopulations even between healthy donors. This inherently weakens the predictive power of covariates based on raw PBMC populations. We observe this phenomenon in our dataset as the combined risk signature had higher predictive power than the baseline one. By opting to present the biological data obtained from the first post-treatment biopsy as change relative to the baseline biopsy, we obtained a much more predictive risk signature in our combined model. This is highlighted in the post-treatment change in CD8CM percentage relative to baseline value which is not only the strongest predictor of PFS in the combined model, but also has a greater univariate predictive value than the whole baseline signature. This validates the potential of our approach has in deconvoluting observed biological phenomena due to patient heterogeneity from that induced by drug treatment, both of which are key to biomarker discovery.

We also validated our risk signature in a second cohort thus allowing us to successfully identify predictive biological biomarkers for therapy across different treatment modalities. The fact that the first trial involved targeted therapies and the second a combination of anti-EGFR and anti-PD-L1 therapy, yet the same blood kinetic change can still distinguish between responders and non-responders is novel and has not been seen before according to our knowledge. The limitation is that the independent validation cohort is small despite the fact that the result we obtained with the present number of validation samples did reach significance (Kandall tau = 0.725, with p-value 0.018). This further highlights the power of multivariate analysis as none of the univariates identified in the test cohort independently predicted treatment outcome in the validation cohort, yet the risk signature strongly correlated with a poorer outcome. We believe that understanding the interplay of key immune subsets between the tumoural microenvironment and the circulation will be key to biomarker development. We focused on two covariates from the risk signature based on their predictive power

in both cohorts. Analysis of immune populations at the tissue level allowed us to identify tissue counterparts of peripheral CD33[+]HLADR[+] monocytes and CD8CM that not only colocalized at the tissue level but also were favourable to a good outcome. Hence, we hypothesize to have uncovered a two-checkpoint system for predicting response to systemic therapy which merits further investigation.

Firstly, a favourable therapeutic outcome requires the presence of high pre-treatment levels of CD14[+]CD16[+]CD33[+]HLADR[high] monocytes. These correlated with a similar population of CD74[+]CD68[+] population of tumoural myeloid cells. Myeloid cells are plastic yet given the similarity in the marker expression patterns between the peripheral CD33[+]CD14[+]CD16[+]HLADR[high] monocyte population and the CD14[+]CD33[+]CD74[+]CD68[+] tissue macrophages, as well as the strong correlation between their corresponding blood and tissue abundance (*Figure 6D*), we hypothesize that they represent the same group of cells that gain CD68 expression as they enter the tissue. This monocyte population matches the population of intermediate monocytes described in the literature shown to have a key role in antigen presentation (*Cravens et al., 2007*; *Zawada et al., 2011*; *Wong et al., 2011*). Given the correlation between tissue levels CD8[+] memory T cell, peripheral intermediate monocytes, and the tissue colocalization of the CD8[+] memory T cells with the CD74[+] macrophages, we can imagine a role for our myeloid population in inducing post-treatment expansion of CD8CM potentially via a mechanism involving cross-presentation of antigens released by drug-induced cancer cell death (*Colbert et al., 2020*). Indeed, cetuximab treatment can be a source of tumour-derived antigens as it has been shown to increase TCR diversity in peripheral T cells as well as cross-presentation by antigen-presenting cells (*Kansy et al., 2018*; *Srivastava et al., 2013*). Ideally, we would require analysis of the draining lymph node to fully understand how these cells move between the tumour and the circulation and their contribution to the expansion of the CD8CM, a role generally accomplished by dendritic cells at the level of the lymph node. Nevertheless, monocyte-derived macrophages have been shown to activate and expand tissue resident memory T cells (Trm) (*Snyder et al., 2021*; *Low et al., 2020*; *Muntjewerff et al., 2020*; *Chu et al., 2020*).

An on-treatment increase in the level of circulating CD8CM significantly correlated with a worse disease outcome in patients who received two regimens of systemic treatments within two trials. Interestingly, high levels of pre-treatment CD8CM cells correlating with a favourable treatment outcome in HNSCC has been previously and independently reported by the Whiteside group indirectly validating our results (*Czystowska et al., 2013*). While an increase in CD8CM correlating with a poorer outcome in both our datasets seems counter-intuitive, it is key to realize that this increase is only seen at the levels of peripheral CD8CM which inversely correlate with tissue Trm (*Figure 6D*). Using single-cell sequencing and mapping of T cell receptor clonality at the levels of TILs and peripheral T cells, researchers have definitively proven that tumour reactive T cell clones can be found in the periphery (*Fairfax et al., 2020*; *Kok et al., 2020*; *Pauken et al., 2021*; *Padrón et al., 2022*). The movement of these cells between the tumour and draining lymph node was recently investigated using photoactivable T cells in mice which showed that there is continuous recruitment of memory T cells from the periphery into the tumour which replenishes the pool of tissue Trm identified by CD103 and TCF1 expression. However, this recruitment is usually matched by an egress of Trm cells out of the tumour to the draining lymph nodes which is accompanied by decreased expression CXCR6 (*Li et al., 2022*). The peripheral origin of Trm cells is still under investigation with recent data suggesting that CD8CM are not only capable of homing into the inflamed tissue but also undergoing transcriptional reprogramming to a stable Trm phenotype (*Matos et al., 2022*). We hypothesize that the predictive power of the population of peripheral CD8CM stems from its ability to home to the tumour and contribute to the Trm population which has recently been identified as key player in anti-tumour immunity in HNSCC as well as other cancers mainly in the context of immunotherapy but also in chemo/radiotherapy (*Han et al., 2021*; *Ida et al., 2021*; *Savas et al., 2018*; *Djenidi et al., 2015*; *Luoma et al., 2022*). Unfortunately, one key shortcoming of our study is the lack of Trm-specific markers such as CD103, CD69, and TCF1 from the imaging mass cytometry panel of antibodies which we believe contributed to this anti-correlation not being stronger.

The accumulation of clonally expanded memory T cells in the periphery in poor responders is the potential result of tumour-induced T cell exclusion which we can infer from the negative correlation observed between peripheral and tumour infiltrating memory T cells (*Jerby-Arnon et al., 2018*). In this model CD8[+] memory T cells are being generated only in patients having a large pre-treatment pool of competent APCs yet these anti-tumour CD8[+] memory cells cannot enter the tumour from the

periphery hence accumulating in circulation. Our group is now focused on understanding the different aspects of this model mainly with regards to the specific tumour exclusion of CD8CM T cells.

Another limitation of our work to discuss is the absence of overall survival (OS) data within our current dataset. It would have been interesting to assess whether the immune markers predict survival in the longer term. However, the accuracy of the predictive signature for OS is often diluted by a variety of subsequent treatment regimens.

The present study shows that the combination of biomarkers established prospectively by liquid biopsies early in the treatment course offers potential for the provision of personalized treatments to patients (*Nenclares et al., 2021*). The risk signature drove the discovery of synergies between the systemic and tumoural immune response. Identifying such complex biological interplay as having a role in disease outcome is impossible without multivariate analysis. The post-stratification survival curves in our study demonstrate markedly different PFS outcomes as a testament to this robust statistical model and could represent an invaluable guide to clinicians during the initial stages of treatment.

## Acknowledgements

We thank the patients who participated in both Phase 2 trials and the staff members at the study sites who cared for them.

The EACH trial was sponsored by University College London and managed by the CRUK and UCL Cancer Trials Centre.

MDF is supported by the UCL/UCLH NIHR Biomedical Research Centre and runs early phase studies in the NIHR UCLH Clinical Research Facility supported by the UCL ECMC.

This research was funded/supported by the National Institute for Health and Care Research (NIHR) Biomedical Research Centre based at Guy's and St Thomas' NHS Foundation Trust and King's College London and/or the NIHR Clinical Research Facility. The views expressed are those of the author(s) and not necessarily those of the NHS, the NIHR, or the Department of Health and Social Care. FUNDING: This work was supported by a grant from Daiichi Sankyo Inc ('Identification of Non-Invasive Treatment Stratification and Longitudinal Monitoring Markers for Patritumab/Cetuximab Combination Therapy'). This work was also supported by Cancer Research UK funding support to King's College London – UCL Comprehensive Cancer Imaging Centre (CR-UK and EPSRC), Cancer Research UK King's Health Partners Centre at King's College London, and Cancer Research UK UCL Centre; University College London (PRB) – Early Detection Award (C7675/A29313); as well as CRUK City of London Centre (CTRQQR-2021\100004). MG, KN, and AAS are supported by Cancer Research UK Clinical Training Fellowships (Award numbers: 163011 for MG, 176885 for KN and 100179 for AAS). LD is supported by EU IMI2 IMMUCAN (Grant agreement number 821558). GA and JMV are supported by CRUK Early Detection and Diagnosis Committee Project grant.

JWO is supported by the UK Medical Research Council (MR/N013700/1) and is a KCL member of the MRC Doctoral Training Partnership in Biomedical Science. FW is also supported by the UK Medical Research Council (MR/N013700/1). JNA is funded by a grant from Cancer Research UK (DCRPGF\100009) and is the recipient of a Cancer Research Institute/Wade FB Thompson CLIP grant (CRI3645).

MTD and KH acknowledge funding support from The Institute of Cancer Research/Royal Marsden Hospital NIHR Biomedical Research Centre and ST acknowledges funding from Guy's and St Thomas' NHS Foundation Trust.

## Additional information

### Competing interests

Paul R Barber: is a shareholder of Nano Clinical Ltd. Kenrick Ng: has received honoraria from Pfizer, GSK/Tesaro and Boheringer Ingleheim, and has had travel/accommodation/expenses paid for by Tesaro. Shahram Kordasti: has received research funding in the form of a grant from Novartis and Celgene. Jana Doyle, Jon Greenberg: is in employment with Daichii Sankyo, and has stock and other ownership

interests, research funding within Daichii Sankyo and has had travel/accommodation/expenses paid for by Daichii Sankyo. Kevin J Harrington: has received honoraria from Amgen; Arch Oncology; Astra-Zeneca; Boehringer-Ingelheim; Bristol-Myers Squibb; Codiak; Inzen; Merck; MSD; Pfizer; Replimune and is on a speakers' bureau for Amgen, AstraZeneca; Bristol-Myers Squibb; Merck, MSD; Pfizer. KH has also received research funding from AstraZeneca, Boehringer-Ingelheim, MSD and Replimune. Martin Forster: has received institutional research funding from AstraZeneca, Boehringer-Ingelheim, Merck and MSD and serves in a consulting or advisory role to Achilles, Astrazeneca, Bayer, Bristol-Myers Squibb, Celgene, Guardant Health, Merck, MSD, Nanobiotix, Novartis, Oxford VacMedix, Pfizer, Roche, Takeda, UltraHuman. Anthony CC Coolen: has stock and other ownership interests with Saddle Point Science Limited. Tony Ng: has received research funding from Astrazeneca and Daichii Sankyo. TN is a founder and shareholder in Nano Clinical Ltd, and PRB is a shareholder. The other authors declare that no competing interests exist.

## Funding

| Funder | Grant reference number | Author |
| --- | --- | --- |
| Cancer Research UK | Early Detection Award C7675/A29313 | Paul R Barber |
| Cancer Research UK | City of London Centre CTRQQR-2021\100004 | Paul R Barber |
| Cancer Research UK | Clinical Fellowship Awards | Kenrick Ng |
| Cancer Research UK | Early Detection and Diagnosis Committee Project grant | Giovanna Alfano Jose M Vicencio |
| Innovative Health Initiative | EU IMI2 IMMUCAN (Grant agreement number 821558) | Luigi Dolcetti |
| Medical Research Council | MR/N013700/1 | James W Opzoomer Felix Wong |
| Cancer Research UK | DCRPGF\100009 | James N Arnold |
| Cancer Research Institute | Wade F.B. Thompson CLIP grant (CRI3645) | James N Arnold |
| Institute of Cancer Research | | Magnus T Dillon |
| Guy's and St Thomas' NHS Foundation Trust | | Selvam Thavaraj |
| Cancer Research UK | | Rami Mustapha Gregory Weitsman Shahram Kordasti |

The funders had no role in study design, data collection and interpretation, or the decision to submit the work for publication.

## Author contributions

Paul R Barber, Conceptualization, Data curation, Software, Formal analysis, Investigation, Methodology, Writing – original draft, Writing – review and editing; Rami Mustapha, Conceptualization, Formal analysis, Methodology, Writing – original draft, Writing – review and editing; Fabian Flores-Borja, Data curation, Formal analysis, Investigation, Methodology; Giovanna Alfano, James W Opzoomer, Data curation, Formal analysis, Investigation; Kenrick Ng, Formal analysis, Methodology, Writing – original draft, Writing – review and editing; Gregory Weitsman, Luigi Dolcetti, Felix Wong, Formal analysis, Investigation; Ali Abdulnabi Suwaidan, Writing – review and editing; Jose M Vicencio, Formal analysis, Investigation, Methodology; Myria Galazi, Data curation, Investigation; James N Arnold, Supervision, Writing – review and editing; Selvam Thavaraj, Methodology; Shahram Kordasti, Formal analysis, Supervision; Jana Doyle, Resources, Supervision, Funding acquisition, Project administration; Jon Greenberg, Resources, Funding acquisition, Project administration, Writing – review and editing; Magnus T Dillon, Methodology, Project administration; Kevin J Harrington, Supervision, Methodology,

Project administration, Writing – review and editing; Martin Forster, Conceptualization, Supervision, Methodology, Writing – review and editing; Anthony CC Coolen, Conceptualization, Data curation, Formal analysis, Supervision, Methodology; Tony Ng, Conceptualization, Resources, Data curation, Supervision, Funding acquisition, Writing – review and editing

## Author ORCIDs
Paul R Barber http://orcid.org/0000-0002-8595-1141
Fabian Flores-Borja http://orcid.org/0000-0002-0881-8822
James W Opzoomer http://orcid.org/0000-0001-6842-756X
Selvam Thavaraj http://orcid.org/0000-0001-5720-7422
Shahram Kordasti http://orcid.org/0000-0002-0347-4207
Anthony CC Coolen http://orcid.org/0000-0002-6976-5875
Tony Ng http://orcid.org/0000-0003-3894-5619

## Ethics
Clinical trial registration NCT02633800.
Written informed consent was obtained for all patients who participated in the Phase 2 clinical trial. Approval was obtained from ethics committees (Research Ethics Committee reference: 15/LO/1670). Approval to procure and process a separate cohort of blood samples from patients at risk of developing lung cancer was also obtained (IRAS ID: 261766).

## Decision letter and Author response
Decision letter https://doi.org/10.7554/eLife.73288.sa1
Author response https://doi.org/10.7554/eLife.73288.sa2

---

# Additional files

## Supplementary files
• Supplementary file 1. Demographic and laboratory-based values of patients in the discovery and validation cohorts, and a comparison of the discovery cohort to the whole parent trial.

• Supplementary file 2. List of covariates for Bayesian multivariate analysis. Twenty-nine laboratory-based covariates were used (obtained at baseline and after one cycle of treatment), and combined with 13 baseline clinical covariates.

• Supplementary file 3. Demographic and laboratory-based values of patients in the discovery cohort separated by arm of treatment on trial (placebo vs. patritumab).

• Supplementary file 4. List of antibodies used in T cell panel and B cell-monocyte panel for analyses of immune cell populations by flow cytometry in the discovery cohort. Antibodies were purchased from BD Biosciences and BioLegend as indicated.

• Supplementary file 5. List of antibodies used in T cell panel and B cell-monocyte panel for analyses of immune cell populations by flow cytometry in the validation cohort. Antibodies were purchased from BD Biosciences and BioLegend as indicated.

• Supplementary file 6. List of antibodies used in mass cytometry (CyToF) imaging analyses for definition of immune cell subpopulations in tissue.

• Supplementary file 7. Risk score signature equations for use with raw covariate values, with missing data imputed with the study mean. These equations were used to calculate the risk score for each patient.

• Transparent reporting form

## Data availability
The data generated in this study and used for multivariate modelling are available from the UCL repository: https://doi.org/10.5522/04/16566207.v1.

The following dataset was generated:

| Author(s) | Year | Dataset title | Dataset URL | Database and Identifier |
|---|---|---|---|---|
| Barber PR, Flores-Borja F, Alfano G, Ng K, Weitsman G, Dolcetti L, Mustapha R, Wong F, Vicencio JM, Galazi M, Opzoomer J, Arnold J, Kordasti S, Doyle J, Greenberg J, Dillon M, Harrington K, Forster MF, Coolen T, Ng T | 2021 | Head and Neck Cancer Multivariate Blood Data | https://doi.org/10.5522/04/16566207.v1 | UCL Research Data Repository, 10.5522/04/16566207.v1 |

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
