## [Editor Report]

While immune checkpoint inhibitors (anti-PD-1 targeted agents) are now FDA-approved for the treatment of locally advanced and recurrent or metastatic head and neck cancer, predictive biomarkers are lacking. In this study, the co-authors have developed an algorithm that they conclude predicts the clinical outcome to multimodality immunotherapy. While this machine learning approach is intriguing, prospective validation of the proposed immune-based signature is essential to begin to incorporate such an approach into the clinic.

---

## [Decision Letter]

**Decision letter after peer review:**

Thank you for submitting your article "Predicting Progression Free Survival after Systemic Therapy in Advanced Head and Neck Cancer: Bayesian regression and Model development" for consideration by *eLife*. Your article has been reviewed by 2 peer reviewers, and the evaluation has been overseen by a Reviewing Editor and Wafik El-Deiry as the Senior Editor. The reviewers have opted to remain anonymous.

*Reviewer #1 (Recommendations for the authors):*

Thinking in terms of what would be clinically tractable, I suggest the authors use ML techniques to reduce the assay panel size to a more parsimonious number of covariates, to train this in their existing dataset, and to then validate this in an independent cohort.

*Reviewer #2 (Recommendations for the authors):*

Barber et al. present a manuscript discussing predictive factors for chemotherapy efficacy in head and neck squamous cancer (HNSCC). The paper is well written, , and its style/formatting are optimal. The baseline signature moderately predicted outcome, and the data after one cycle further improved the algorithm, though this decreases its utility as a pure predictive tool. It is interesting that a subpopulation of monocytes, a subset of white peripheral cells long suspected to correlate with outcomes in HNSCC was one of the key drivers of the algorithm. However the overall impact in the field of this work seems limited.

Comments:

– The authors focused on immune cell subpopulations and exosomes, which narrows the scope (no cytokines or other biomarkers were included).

– The signatures were not prospectively validated on an independent cohort.

– Unfortunately this algorithm predicts outcome for a first-line therapy that is no longer considered to be the standard of care for HNSCC.

– The outcome measure is PFS, which is appropriate for therapy effect but not the standard in first line therapy (OS would be).

– The conclusions of the manuscript are supported by the data, but some of the caveats (such as the lack of a validation cohort, key in predictive biomarker development), are not addressed.

[Editors' note: further revisions were suggested prior to acceptance, as described below.]

Thank you for resubmitting the paper entitled "Predicting Progression Free Survival after Systemic Therapy in Advanced Head and Neck Cancer: Bayesian regression and Model development" for further consideration by *eLife*. Your revised article has been evaluated by a Senior Editor and a Reviewing Editor. We are sorry to say that we have decided that this submission will not be considered further for publication by *eLife*.

*Reviewer #1 (Recommendations for the authors):*

Thank you for the opportunity to review this revised manuscript. I feel that the authors have made a good-faith effort to answer all of my prior comments. I, unfortunately, feel that the prior comments have not been very completely addressed.

First is the issue of unmet clinical need. Several reviewers asked whether a predictive biomarker for chemotherapy is relevant, in the era of ICB. I do in fact agree with the authors that there is still clinical value to predicting response to regimens such as EXTREME which do in fact still have a role in the treatment of HNSCC, and are understudied.

However, I did raise the question as to whether this fairly labor-intensive, difficult-to-scale, and expensive assay would add value to the fairly inexpensive and widely available (although admittedly not widely used) biomarkers we already have. I gave the example of NLR but also alluded to simple nomograms that include clinical factors such as age, performance status, tumor stage, etc. It seems important to more convincingly make the case that this model outperforms the widely available, inexpensive data we already have. Looking at the c-indices for this model, I am not sure this is the case. I again would advise this be shown rigorously with direct comparisons, to make the case that this model adds value.

The validation set is very important to the conclusion that this assay has potential clinical use. I fully recognize that a validation set is hard to come by. But this is the "acid test" for such an assay. The validation data here is promising, and again, I understand how hard it is to gather such data, but it is not very convincing. Only 8 patients, with no PFS data (iRECIST is used, with a 50% objective response rate, which seems very high). To support the claim that such an assay has clinical value, a much broader set of validation data are needed.

I raised the question earlier about the large number of patients excluded due to a lack of biospecimens and am concerned that this may limit the results, as this is akin to unequal censoring in a clinical trial. It would be important to compare the patients being studied and those not being studied. This was not addressed.

As with the other reviewers, the lack of OS data is a weakness here. This is a more minor point, as I do understand that the reviewers tried and were unable to obtain this data from the industry sponsor. I want to stress this is a minor weakness compared to some of the above points. However, multiple co-authors have very significant industry COI (co-authors each receiving personal payments from 15-20 companies) and I am concerned that the withholding of OS data (which has already been collected and is in fact a component of PFS) may reflect other motives on the part of the industry sponsor with whom some authors may be conflicted. Nevertheless, if the data are unavailable, a suggestion: if OS data are not available, perhaps the authors could show rigorously that PFS is a very good surrogate marker of OS in this disease.

---

## [Author Response]

Reviewer #2 (Recommendations for the authors):Barber et al. present a manuscript discussing predictive factors for chemotherapy efficacy in head and neck squamous cancer (HNSCC). The paper is well written, , and its style/formatting are optimal. The baseline signature moderately predicted outcome, and the data after one cycle further improved the algorithm, though this decreases its utility as a pure predictive tool. It is interesting that a subpopulation of monocytes, a subset of white peripheral cells long suspected to correlate with outcomes in HNSCC was one of the key drivers of the algorithm. However the overall impact in the field of this work seems limited.Comments:– The authors focused on immune cell subpopulations and exosomes, which narrows the scope (no cytokines or other biomarkers were included).

Thank you. We selected a finite number of covariates based on a few factors – (a) published literature, (b) previous data generated by the group and (c) the applicability of the findings to the clinic. Instead of an exploratory article in which we could generate an infinite number of covariates by a technique similar to RNA sequencing, we opted for a select set of covariates. This hypothesis-driven approach generated a strong signature that is now validated across two trials. The focus on immune population is driven by our hypothesis that systemic changes in the PBMCs are indicative and reflective of the status of the intra-tumoral immune response. In the revised manuscript we used a custom immune focused imaging mass cytometry antibody panel to probe tissue sections from 9 patients. We now show that the key populations driving the predictive model in the periphery are not only reflected at the tumoral level, but these disparate immune cell subpopulations also interact. Figure 6 from the manuscript is shown below, in which we use a machine learning approach to segment cells and assign them to distinct immunological subpopulations. We found that the peripheral monocyte population strongly correlated with a tumoral macrophage population having a similar marker expression pattern. We found that the peripheral central memory CD8 T cells inversely correlated with tissue resident memory T cells. The tissue presence of both these cells correlated positively with outcome. Most strikingly, these two populations were most likely to co-localize with each other at the tissue level at a frequency of almost double the second highest co-localization. Data on the nature of the interplay between peripheral systemic immunity and intra-tumoral immunity is novel and rarely exists in the literature outside the scope of in-vivo animal models. Here we describe these interactions using human patient samples treated with a clinically relevant therapy.

Given the limited amount of patient sera collected in the trial we opted to perform exosome analysis on markers known to impact the response to the anti-EGFR/HER3 treatment/immune responses. This was in line with our labs work to use exosome FRET-FLIM as a surrogate for tissue FRET-FLIM which we originally used to discover a potential dimer dependent mechanism for anti-EGFR treatment resistance in neoadjuvant breast cancer patients^9^; and more recently published on a colorectal patient sample cohort from the COIN study ^10^. While exosome EGFR-HER3 heterodimer failed to reach significance in our risk signature, it was close as depicted in the Kaplan-Meier curve from Figure 3C shown below. We of course acknowledge the potential added benefit of having serum cytokine array analysis. While that was not feasible for this study our group now aims at ensuring that extra patient serum samples are bio-banked for such analysis from ongoing and future trials.

– The signatures were not prospectively validated on an independent cohort.

We completely agree with the reviewer regarding the need to obtain a validation set. Obtaining patient samples from a similar cohort was difficult but we managed to validate the signature on a set of patients treated with an anti-PDL1 monoclonal antibody in combination with Cetuximab. Furthermore, the validation was performed using a limited numbers of covariates that were identified in the risk signature by the Bayesian model. These immune populations can be obtained by running a limited set of markers on flow cytometry. We were very happy to see that these limited immune based covariates strongly correlated with a worst disease response in an independent cohort using a different treatment modality. This furthers our hypothesis that changes in the immune populations are key to understanding response to systemic therapy. Fueled with the data from the validation cohort we furthered our analysis of the tissue from a total of 9 patients from the test cohort. Using imaging mass cytometry, we were able to identify how immune populations are mirrored at the tumoral level opening the horizon for new research. The data for the validation set is copied into this letter in response to point 2 of the public evaluation summary.

– Unfortunately this algorithm predicts outcome for a first-line therapy that is no longer considered to be the standard of care for HNSCC.

The EXTREME regimen (platinum/5-FU/cetuximab) remains a first-line standard of care treatment in the UK and European countries for HNSCC patients with negative PDL1 status (CPS score <1) which account for around 15% of all HNSCC patients ^7^. While the US Food and Drug Administration (FDA) approved pembrolizumab in combination with chemotherapy as first-line treatment regardless of PD-L1 expression and pembrolizumab alone for patients with PD-L1-expressing tumours (CPS ≥1), the European Medicines Agency (EMA) approved pembrolizumab with or without chemotherapy only for patients with a CPS ≥1, and this has been highlighted in the European Society for Medical Oncology (ESMO) and the UK National Institute for Health and Care Excellence (NICE) guidelines ^8^ and (https://www.nice.org.uk/guidance/ta661/chapter/1-Recommendations).

Furthermore, chemotherapy with EXTREME regimen is standard of care for patients with contraindications to immune checkpoint inhibitors such as autoimmune disease ^8^. It can also be considered as second-line treatment in patients who only received pembrolizumab monotherapy in the first line setting.

– The outcome measure is PFS, which is appropriate for therapy effect but not the standard in first line therapy (OS would be).

Thank you. We admit that the absence of overall survival is an inherent limitation of the study. In the process of submitting this revision, we have once again requested this dataset from the sponsoring pharmaceutical company but were informed that they are unable to provide it. This is because reorganization of funding priorities within the company precludes them opening datasets from an already-published clinical trial. We are equally disappointed to not be able to obtain this data, but firmly believe that the ability of the signature to predict PFS (the primary endpoint of the trial, untainted by subsequent lines of treatment), as well as cross-validation against the contemporary EACH trial, is a testament to the signature’s strength.

– The conclusions of the manuscript are supported by the data, but some of the caveats (such as the lack of a validation cohort, key in predictive biomarker development), are not addressed.

We hope that the reviewer accepts the revised manuscript in which we believe we have addressed all his comments.

References

1. Padron, L. J. *et al.* Sotigalimab and/or nivolumab with chemotherapy in first-line metastatic pancreatic cancer: clinical and immunologic analyses from the randomized phase 2 PRINCE trial. *Nat Med* 28, 1167-1177, doi:10.1038/s41591-022-01829-9 (2022).

2. Luoma, A. M. *et al.* Tissue-resident memory and circulating T cells are early responders to pre-surgical cancer immunotherapy. *Cell*, doi:10.1016/j.cell.2022.06.018 (2022).

3. Hood, S. P. *et al.* Identifying prostate cancer and its clinical risk in asymptomatic men using machine learning of high dimensional peripheral blood flow cytometric natural killer cell subset phenotyping data. *ELife* 9, doi:10.7554/*eLife*.50936 (2020).

4. Shen, R. *et al.* LAG-3 expression on peripheral blood cells identifies patients with poorer outcomes after immune checkpoint blockade. *Sci Transl Med* 13, doi:10.1126/scitranslmed.abf5107 (2021).

5. Czystowska, M. *et al.* The immune signature of CD8(+)CCR7(+) T cells in the peripheral circulation associates with disease recurrence in patients with HNSCC. *Clin Cancer Res* 19, 889-899, doi:10.1158/1078-0432.CCR-12-2191 (2013).

6. Krieg, C. *et al.* High-dimensional single-cell analysis predicts response to anti-PD-1 immunotherapy. *Nat Med* 24, 144-153, doi:10.1038/nm.4466 (2018).

7. Burtness, B. *et al.* Pembrolizumab alone or with chemotherapy versus cetuximab with chemotherapy for recurrent or metastatic squamous cell carcinoma of the head and neck (KEYNOTE-048): a randomised, open-label, phase 3 study. *Lancet* 394, 1915-1928, doi:10.1016/S0140-6736(19)32591-7 (2019).

8. Machiels, J. P. *et al.* Squamous cell carcinoma of the oral cavity, larynx, oropharynx and hypopharynx: EHNS-ESMO-ESTRO Clinical Practice Guidelines for diagnosis, treatment and follow-up. *Ann Oncol* 31, 1462-1475, doi:10.1016/j.annonc.2020.07.011 (2020).

9. Tao, J. J. *et al.* Antagonism of EGFR and HER3 enhances the response to inhibitors of the PI3K-Akt pathway in triple-negative breast cancer. *Sci Signal* 7, ra29, doi:10.1126/scisignal.2005125 (2014).

10 .Barber, P. R. *et al.* HER2-HER3 Heterodimer Quantification by FRET-FLIM and Patient Subclass Analysis of the COIN Colorectal Trial. *J Natl Cancer Inst* 112, 944-954, doi:10.1093/jnci/djz231 (2020).

11. Mascarella, M. A., Mannard, E., Silva, S. D. & Zeitouni, A. Neutrophil-to-lymphocyte ratio in head and neck cancer prognosis: A systematic review and meta-analysis. *Head Neck* 40, 1091-1100, doi:10.1002/hed.25075 (2018).

12. Takenaka, Y. *et al.* Prognostic role of neutrophil-to-lymphocyte ratio in head and neck cancer: A meta-analysis. *Head Neck* 40, 647-655, doi:10.1002/hed.24986 (2018).

13. Ju, J. *et al.* Nomograms predicting long-term overall survival and cancer-specific survival in head and neck squamous cell carcinoma patients. *Oncotarget* 7, 51059-51068, doi:10.18632/oncotarget.10595 (2016).

14. Sano, D. *et al.* Real-world Treatment Outcomes of the EXTREME Regimen as First-line Therapy for Recurrent/Metastatic Squamous Cell Carcinoma of the Head and Neck: A Multi-center Retrospective Cohort Study in Japan. *Anticancer Res* 39, 6819-6827, doi:10.21873/anticanres.13898 (2019).

15. Guigay, J. *et al.* Observational, prospective, phase 4 study in patients with first-line recurrent and/or metastatic squamous cell carcinoma of the head and neck treated with cetuximab and platinum-based therapy: DIRECT. *Cancer Rep (Hoboken)* 5, e1467, doi:10.1002/cnr2.1467 (2022).

[Editors' note: further revisions were suggested prior to acceptance, as described below.]

Reviewer #1 (Recommendations for the authors):Thank you for the opportunity to review this revised manuscript. I feel that the authors have made a good-faith effort to answer all of my prior comments. I, unfortunately, feel that the prior comments have not been very completely addressed.

We would like to thank the reviewer for taking the time to go through our revisions and the detailed rebuttal letter. However, we are sorry that he felt that our revision did not address all his concerns. We were confident and reasonably hopeful that the inclusion of a difficult to obtain validation cohort and the significant expansion of our immunological data would be sufficient to satisfy the previous reviewers’ comments. The fact that the first trial involved targeted therapies and the second a combination of anti-EGFR and anti-PDL1 therapy, yet the same blood kinetic change can still distinguish between responders and non-responders is very novel and unprecedented according to our knowledge and reinforced the originality of the research. We also regret that the immune cell biology findings from the deep profiling of our two Phase 2 studies were largely ignored despite their implications for the development of future immunotherapeutic. For instance, the additional imaging CyTOF data (in Figure 6) showed a very novel finding that the two most significant immune cell subpopulations (CD33+CD14+HLADRhigh monocytes and CD8 central memory T cells) were most likely to co-localize with each other at the tissue level at a frequency of almost double the second highest co-localization. This suggests that in-situ antigen presentation is a key driver for therapy induced response across multiple therapeutic modalities (e.g. standard of care chemotherapy and targeted therapies). These results once published will stimulate the immune-oncology community to further test this specific cell-cell interaction mechanism in preclinical mouse model treated e.g. with immune checkpoint therapeutics.

First is the issue of unmet clinical need. Several reviewers asked whether a predictive biomarker for chemotherapy is relevant, in the era of ICB. I do in fact agree with the authors that there is still clinical value to predicting response to regimens such as EXTREME which do in fact still have a role in the treatment of HNSCC, and are understudied.However, I did raise the question as to whether this fairly labor-intensive, difficult-to-scale, and expensive assay would add value to the fairly inexpensive and widely available (although admittedly not widely used) biomarkers we already have. I gave the example of NLR but also alluded to simple nomograms that include clinical factors such as age, performance status, tumor stage, etc. It seems important to more convincingly make the case that this model outperforms the widely available, inexpensive data we already have. Looking at the c-indices for this model, I am not sure this is the case. I again would advise this be shown rigorously with direct comparisons, to make the case that this model adds value.

We strongly disagree with the “no added value” comment. As a translational research group, we have been passionate about applying modern omic technology coupled with mathematical modelling and machine learning to Phase 1/Phase 2/Phase 3 clinical trial samples (Barber, Weitsman et al., 2020), with a goal of gaining a deeper understanding of the scientific mechanisms behind heterogeneity of response between patients to targeted and/or immune-based therapeutics. Also I sincerely believe that this early phase trial related translational work, underpinned by deep profiling techniques (rather than the crude neutrophil:lymphocyte ratio (NLR)) has shown us the way to find hidden subpopulations of patients who may be able to respond better; and for those the models predict will be resistant to treatment, we can in the future offer them new treatments based on the molecular immunological information about the individual patients arising from these deep profiling techniques.

As an illustration, our results echo the recent finding that treatment-induced expansion of emergent T cell clones in tumors and the systemic circulation that were undetectable prior to immune checkpoint blockade therapy (Luoma, Suo et al., 2022). The latter requires looking at multiple time points and is similar to our findings that it is the analysis of the blood kinetic changes between pretreatment and (early) post treatment time points that reveals the memory CD8^+^ T cell expansion which predict response. These memory T cells are now believed to be bona fide responders to PD-1/PD-L1 blockade according to recent results obtained from preclinical models (Huang, Wu et al., 2022).

It was precisely the deployment of deep profiling (Spectral flow cytometry analysis (Luoma et al., 2022)) that leads to the conclusion of this recent translational trial paper (which unlike ours does not have an independent phase 2 trial to confirm the findings). We therefore strongly disagree with the reviewer’s criticism on the basis of the lack of need for expensive assays (compared to NLR) as a reason to reject our revised work.

We also believe that the value of this work should not be hindered by its ability to be readily implemented in the clinic by directly comparing it to clinical nomograms. While we appreciate the relevance and importance of generic clinical nomograms, they often lack the specific type of data required for discovery and development of next generation precision and personalized medicine, by identifying future targets for overcoming resistance, for instance.

Summing up, our work is the first of its kind where the derived deep profiling signature was cross validated in a separate independent cohort, and the drug induced changes in immune populations were assessed at the tumoral and systemic level furthering our understanding of immune cells interaction and trafficking.

The validation set is very important to the conclusion that this assay has potential clinical use. I fully recognize that a validation set is hard to come by. But this is the "acid test" for such an assay. The validation data here is promising, and again, I understand how hard it is to gather such data, but it is not very convincing. Only 8 patients, with no PFS data (iRECIST is used, with a 50% objective response rate, which seems very high). To support the claim that such an assay has clinical value, a much broader set of validation data are needed.

Regarding the small sample size criticism, our collaborator Prof. Ton Coolen is one of the few statistical physics/machine learning experts (https://www.ru.nl/en/people/coolen-a) who was invaluable in helping us to review a recent ML paper https://doi.org/10.7554/*eLife*.80150, which would have been difficult to review without his specialist opinion. The following is his response to this specific Reviewer’s comment on our paper:

“Regarding the small number of patients in the validation cohort; there was an a priori chance that this cohort would not have demonstrated the correlation of the risk score with treatment response, and that more patients would have been required. However, the result we obtained with the present number of validation samples (Kandall tau = 0.725, with p-value 0.018) indicates that the number of validation samples suffices. The p-value, by definition, already takes into account the number of patients in a cohort — for smaller data sets it is easier to observe correlations as a result of fluke, and p-values are typically larger. Our present result states that the likelihood of achieving for the validation data presently used a correlation value of 0.725 or higher by chance, i.e. under the null hypothesis of statistical independence, is 0.018. This being below the standard confidence level of 0.05 for statistical significance, one must conclude with confidence that our validation data is correlated with the risk score, and hence no further patients are necessary to support this claim.”

Hence obtaining a significant correlation despite the small sample size in fact empowers our model.

The 50% objective response rate in this validation cohort is rather a reflection of improved response and efficacy of combining the anti-EGFR and anti-PDL1 in this setting. This is based on the preliminary finding of the trial which was presented at the European Society for Medical Oncology (ESMO) Annual Congress in 2020 and published as an abstract (M. Forster, 2020).

I raised the question earlier about the large number of patients excluded due to a lack of biospecimens and am concerned that this may limit the results, as this is akin to unequal censoring in a clinical trial. It would be important to compare the patients being studied and those not being studied. This was not addressed.

We understand the reviewer’s concern with regards to a potential skewing of outcome data due to some patient exclusions. While we attempted to address this concern by showing no difference in PFS when comparing our study to the literature, we accept that this answer did not meet the reviewer’s expectations. Hence in this revised manuscript we have amended supplementary file 1 to compare the patient characteristics between the discovery sub-cohort assessed in this study with the original study cohort (Dillon, Grove et al., 2019). The table clearly shows that the clinical characteristics of the included sub-cohort were similar to those in the published full cohort. Hopefully this comparison will convince the reviewer that the data is robust and not skewed by patient subsampling due to availability of biological specimens.

As with the other reviewers, the lack of OS data is a weakness here. This is a more minor point, as I do understand that the reviewers tried and were unable to obtain this data from the industry sponsor. I want to stress this is a minor weakness compared to some of the above points. However, multiple co-authors have very significant industry COI (co-authors each receiving personal payments from 15-20 companies) and I am concerned that the withholding of OS data (which has already been collected and is in fact a component of PFS) may reflect other motives on the part of the industry sponsor with whom some authors may be conflicted. Nevertheless, if the data are unavailable, a suggestion: if OS data are not available, perhaps the authors could show rigorously that PFS is a very good surrogate marker of OS in this disease.

We do regret that the OS data is not available, and we thank the reviewer for acknowledging that this is a minor point. We have previously explained the reason for the lack of OS data in this work due to the reorganization of funding priorities within the sponsor which precluded them from accessing datasets from an already-published clinical trials. We are equally disappointed to not be able to obtain this data, but firmly believe that the ability of the signature to predict PFS (the primary endpoint of the trial, untainted by subsequent lines of treatment), as well as cross-validation against the contemporary EACH trial, is a testament to the signature’s strength.

The PFS has been extensively studied across many cancer types as a surrogate for overall survival in oncology trials (cite: https://www.nature.com/articles/s41416-020-0805-y) and it was a valid surrogate end point for OS to assess treatment effect of chemotherapy in loco-regionally advanced nasopharyngeal carcinoma (PFS was strongly correlated with OS at the individual level (ρ = 0.93, 95% confidence interval [CI] = 0.93 to 0.94)) and at the trial level (R2 = 0.95, 95% CI = 0.47 to 1.00) (cite: https://academic.oup.com/jnci/article/109/4/djw239/2605763?login=false). A similar high correlation between PFS and OS was also observed in HNSCC in response to PD1/PDL1 immune checkpoint inhibitors (in the ITT population (rho = 0.76) and PD-L1-positive population (rho = 0.74)) (Ye, Ji et al., 2020).

References

Barber PR, Weitsman G, Lawler K, Barrett JE, Rowley M, Rodriguez-Justo M, Fisher D, Gao F, Tullis IDC, Deng J, Brown L, Kaplan R, Hochhauser D, Adams R, Maughan TS, Vojnovic B, Coolen ACC, Ng T (2020) HER2-HER3 Heterodimer Quantification by FRET-FLIM and Patient Subclass Analysis of the COIN Colorectal Trial. J Natl Cancer Inst 112: 944-954

Dillon MT, Grove L, Newbold KL, Shaw H, Brown NF, Mendell J, Chen S, Beckman RA, Jennings A, Ricamara M, Greenberg J, Forster M, Harrington KJ (2019) Patritumab with Cetuximab plus Platinum-Containing Therapy in Recurrent or Metastatic Squamous Cell Carcinoma of the Head and Neck: An Open-Label, Phase Ib Study. Clin Cancer Res 25: 487-495

Huang Q, Wu X, Wang Z, Chen X, Wang L, Lu Y, Xiong D, Liu Q, Tian Y, Lin H, Guo J, Wen S, Dong W, Yang X, Yuan Y, Yue Z, Lei S, Wu Q, Ran L, Xie L et al. (2022) The primordial differentiation of tumor-specific memory CD8(+) T cells as bona fide responders to PD-1/PD-L1 blockade in draining lymph nodes. Cell 185: 4049-4066 e25

Luoma AM, Suo S, Wang Y, Gunasti L, Porter CBM, Nabilsi N, Tadros J, Ferretti AP, Liao S, Gurer C, Chen YH, Criscitiello S, Ricker CA, Dionne D, Rozenblatt-Rosen O, Uppaluri R, Haddad RI, Ashenberg O, Regev A, Van Allen EM et al. (2022) Tissue-resident memory and circulating T cells are early responders to pre-surgical cancer immunotherapy. Cell 185: 2918-2935 e29

M. Forster RM, J. Sacco, A. Kong, G. Wheeler, S. Forsyth, R. Bhat, K. Blair, J. Ward, H. Lowe, V. Spanswick, L. Ensell, J. Hartley, L. White (2020) 922P – EACH: A phase II study evaluating the safety and anti-tumour activity of avelumab and cetuximab in recurrent/metastatic squamous cell carcinomas. In ESMO

Ye J, Ji X, Dennis PA, Abdullah H, Mukhopadhyay P (2020) Relationship Between Progression-Free Survival, Objective Response Rate, and Overall Survival in Clinical Trials of PD-1/PD-L1 Immune Checkpoint Blockade: A Meta-Analysis. Clin Pharmacol Ther 108: 1274-1288